# New Strategy for the Design of Anti-Corrosion Coatings in Bipolar Plates Based on Hybrid Organic–Inorganic Layers

**DOI:** 10.3390/molecules28073279

**Published:** 2023-04-06

**Authors:** Xiaoxuan Li, Wenjie Sun, Yuhui Zheng, Chenggang Long, Qianming Wang

**Affiliations:** 1Guangzhou Key Laboratory of Analytical Chemistry for Biomedicine, School of Chemistry, South China Normal University, Guangzhou 510006, China; 2Ruide Technologies (Foshan) Inc., Foshan 528311, China

**Keywords:** polypyrrole, inorganic, anti-corrosion, hybrid layer

## Abstract

As a star material in conducting polymers, a polypyrrole coating was assembled onto the surface of 316 stainless steel by an electrochemical method. In the next step, the composite layer consisting of carbon nitride nanosheets (CNNS) and polymethyl methacrylate (PMMA) was sprayed. The corrosion manner of composite coatings in a simulated proton-exchange membrane fuel cell (PEMFC) environment was evaluated. The results show that the final coating generated at a voltage of 1.0 has demonstrated the optimized corrosion resistance. The polypyrrole layer improves the corrosion resistance of the stainless steel substrate, and the CNNS/PMMA coating further strengthens the physical barrier effect of the coating in corrosive solutions.

## 1. Introduction

The long-standing popularity of proton-exchange membrane fuel cells (PEMFCs) has developed the most advanced, cleanest and most efficient energy-generating equipment for future application. The basic principle lies in the direct conversion of hydrogen and oxygen into electrical energy and only produces water as a byproduct [1]. At the same time, due to its high power density, reduced pollution emissions, low working temperature and great portability, it is expected to be widely used in transportation and household power supply. A PEMFC is mainly composed of a proton-exchange membrane, a catalyst layer, a gas diffusion layer and bipolar plates [2]. The gas diffusion layer, the catalyst layer and the polymer electrolyte membrane are prepared by hot pressing to obtain a membrane electrode assembly (MEA). The proton-exchange membrane in the middle plays multiple roles in conducting protons (H^+^), preventing electron transfer and isolating the cathode and anode reactions. The catalyst layers on both sides are the places where fuel and oxidant have electrochemical reactions. The main function of the gas diffusion layer is to support the catalyst layer, stabilize the electrode structure, provide a gas transmission channel and improve water management.

It has to be emphasized that the main function of bipolar plates is to separate the reaction gas, introduce the reaction gas into the fuel cell through the flow field, collect and conduct the current, support the membrane electrode, and undertake the heat dissipation and water drainage functions of the whole fuel cell.

Bipolar plates are among the most important components in PEMFCs, accounting for approximately 80% of the total weight [3] and 40% of the cost [4]. At present, bipolar plates are usually made of graphite and metal. However, the porous structure, low mechanic strength, high permeability and brittleness of graphite severely hamper the power density of fuel cell stack, and the high cost of graphite hinders further large-scale commercialization. Therefore, metal bipolar plates are more widely used in business because of their high conductivity and mechanical strength, low permeability and low cost. Moreover, metal materials can be processed into thin plates with a simple treatment, which greatly reduces the size and weight of PEMFCs. It has been well accepted that stainless steel can be easily available and promising for metal bipolar plates. However, the main disadvantage of stainless steel is that it is easily corroded under working condition of PEMFCs. Then, dissolved ions, such as Fe, Cr and Ni ions from the stainless steel matrix, may contaminate the Pt catalyst and the membrane electrodes [5].

An effective way to solve the conductivity and corrosion resistance of bipolar plates is to modify the metal surface coating [6]. A great deal of research work has been performed on the surface modification coating technology of metal bipolar plates of fuel cells at home and abroad. Commonly used preparation methods of modified coatings include chemical vapor deposition [7], physical vapor deposition [8], closed field unbalanced magnetron sputter ion plating [9] and high-energy micro-arc alloying [10].

However, these approaches often require expensive instruments and equipment. In recent years, numerous teams contributed to the search for new ways of improving the metal surface at low cost. For metal bipolar plates, a feasible and effective solution is to coat them with a conductive anti-corrosive coating. Conductive polymer (CP) coatings [11], such as polypyrrole (pPy), polyaniline (PAni) and polythiophene (PTh) coatings, are ideal potential candidate materials due to their unique advantages in conductivity, electrochemical performance, mechanical strength, and chemical and electrochemical synthesis.

I.Martins [12] electrochemically synthesized the pPy coating on the surface of galvanized steel in tartaric acid aqueous solution, and conducted electrochemical tests on bare steel, bare zinc, galvanized steel and galvanized steel with a pPy coating in NaCl, HCl and H_2_SO_4_ solutions. The results revealed that pPy can be used as a protective coating to improve the corrosion resistance of metal. Simultaneously, the existence of a conductive pPy coating can not only perform a predominant role in mechanical isolation, but can also repair the passivation coating on the metal surface, thus reducing the corrosion rate. After the conductive polymer is coated on the steel surface, the corrosion potential of the steel will be greatly increased, which makes the corrosion less likely to occur. T.J.Pan [13] electrodeposited a pPy/graphene composite coating on the surface of 304SS by cyclic voltammetry, studied the corrosion of the composite material in a proton-exchange membrane fuel cell environment, and compared it with a single pPy coating and substrate. The composite coating improved the corrosion resistance of the substrate more effectively than a single pPy coating, because it reduced the corrosion current density of the substrate and kept a high open circuit potential during the whole soaking process. In addition, the composite coating has higher chemical stability and better conductivity than the single pPy coating. This is mainly due to the structure of composite materials containing graphene, which has good electrical conductivity and corrosion resistance.

Liu successfully electrodeposited a conductive pPy/C-PDA coating on the surface of 304SS by introducing dopamine-functionalized carbon powder [14]. Compared with the original pPy/C coating, the pPy/C-PDA coating has smoother and more uniform surface, better thermal stability and lower contact resistance. Specifically, the pPy/C-PDA coating has a long-term corrosion resistance of 720 h in the simulated proton-exchange membrane fuel cell environment.

Polymethyl methacrylate (PMMA), also known as acrylic or plexiglass, has high transparency, a low price and easy machining, and has been used to prepare corrosion-resistant coatings [15,16,17,18]. Recent studies concern the synthesis of organic–inorganic hybrid materials of CeO_2_/Ce_2_O_3_ nanoparticles covalently bonded to PMMA by the sol-gel method, especially the long-term corrosion resistance has been explored [16,17].

With the aim of improving the anti-corrosion performance of metal plates, we electrodeposited a pPy coating and a pPy/C coating on the surface of 316SS by a potentiostatic method, then mixed carbon nitride nanosheets and PMMA solution, and sprayed the mixed solution on the surface of the pPy/C coating to prepare corrosion-resistant coatings. The corrosion resistance of these coatings in the simulated proton-exchange membrane fuel cell environment has been studied by an electrochemical test including open circuit potential (E_ocp_) versus time, an electrochemical polarization curve, and electrochemical impedance spectroscopy (EIS). The interfacial contact resistance (ICR) of the different composite coatings was measured to evaluate its conductivity. Through our research, we found that CNNS/PMMA/pPy/C-1.00 demonstrated optimal corrosion resistance as a corrosion-resistant coating.

## 2. Experimental Sections

### 2.1. Materials and Reagents

The 316SS sheet was selected as the substrate in this study. The steel sheet was mounted in epoxy resin with an exposed surface area of 2 × 2 cm^2^. The steel sheet was polished mechanically to 1500 mesh, then cleaned with ethanol and distilled water, and finally dried in vacuum. Graphite powder was provided by the Ruide Technologies (Foshan) Inc. Foshan, Guangdongi, China. The pyrrole was purchased from Macklin Inc, Shanghai, China. Oxalic acid (H_2_C_2_O_4_), poly(methyl methacrylate) (PMMA), melamine, sulfuric acid (H_2_SO_4_), and hydrofluoric acid (HF, 40%) were purchased from Aladdin biological technology Co., Ltd., Shanghai, China. All the reagents were used without any purification.

### 2.2. Electropolymerization of pPy-316 and pPy/C-316 Coatings

The pPy and Ppy/C coatings were completed using a potentiostatic method on a CHI660E electrochemical workstation. A typical three-electrode device was established: 316SS was used as the working electrode, and a platinum electrode and a saturated calomel electrode (SCE) were used as the counter electrode and reference electrode, respectively.

Single Ppy coatings were synthesized on a 316SS plate in an aqueous electrolyte solution of 0.1 M pyrrole and 0.3 M oxalic acid using a potentiostatic technique, and the electrolyte was purged with nitrogen for 30 min before use to remove any dissolved oxygen. Then, pyrrole was electrochemically polymerized and deposited on the surface of 316SS in the potentiostatic mode, and Ppy-modified 316SS was obtained. The electropolymerization process was conducted under continuous stirring in the ice-water bath. After 10 min scanning, the samples were washed with anhydrous ethanol and distilled water, and dried in the air. By changing the constant potential value, the Ppy samples prepared under different constant potential conditions were successively obtained, and labeled Ppy-0.80, Ppy-0.90, Ppy-1.00 and Ppy-1.10.

As a control sample, the Ppy/C composite coatings were produced from the same electrolyte solution in the presence of 1 g/L graphite powder, and electrodeposited on 316SS using a similar electrochemical synthesis procedure. After the addition of graphite powder to the electrolyte solution, an ultrasonic dispersion treatment was used for 1 h to ensure the complete dispersion of the graphite powder in the electrolyte solution. By changing the constant potential value, the Ppy samples prepared under different constant potential conditions were successively obtained, and labeled Ppy/C-0.80, Ppy/C-0.90, Ppy/C-1.00 and Ppy/C-1.10.

### 2.3. Synthesis of CNNS/PMMA/Ppy-316 and CNNS/PMMA/Ppy/C-316 Coatings

The carbon nitride nanosheets were prepared by a bottom-up method as follows: A certain amount of melamine was transferred into a tube furnace. The temperature of the tubular furnace was increased to 550 °C at a heating rate of 10 °C·min^−1^, and then the sample was subjected to 550 °C for 2 h. After heat treatment, the furnace was cooled down naturally to room temperature with continuous N2 flowing.

As for the preparation of a dichloromethane solution of 1% PMMA, the solution was stirred until PMMA was completely dissolved. The same 1% methylene chloride suspension of carbon nitride nanosheets was prepared by this method. After the suspension was stirred for 30 min, an ultrasonic dispersion treatment was used for 10 min to ensure complete dispersion of the carbon nitride nanosheets in the suspension. According to a certain proportion, the above two solutions were employed and the ultrasonic treatment was performed for 1 h. The obtained suspension was added into a spray gun, wherein the nozzle of the spray gun was parallel to the surface of the substrate with a distance of 15–20 cm, and spraying the substrate. During the spraying process, the nozzle was kept parallel to the substrate, and the moving speed was moderate and constant, so as to ensure that the suspension was completely and uniformly sprayed on the substrate surface. After spraying, the samples were dried and solidified at 80 °C, then the specimen was taken out for natural cooling. The sample sprayed with composite material on the surface of the ppy film was named CNNS/PMMA/Ppy, and the sample sprayed with composite material on the surface of the ppy film was named CNNS/PMMA/Ppy/C. Samples directly sprayed with composite material on the 316SS surface are recorded as CNNS/PMMA. It should be noted that in order to refine the experimental results, additional samples were prepared in the meantime, and labeled as CNNS/PMMA/Ppy/C-0.95 and CNNS/PMMA/Ppy/C-1.05, respectively.

### 2.4. Characterization

The microstructural information was measured by a scanning electron microscope (FE-SEM FEI Quanta 250FEG). Powder X-ray diffraction (XRD) was analyzed by a Rigaku MiniFlex600 diffractometer. The degree of graphitizing of graphite powder was studied with a RENISHAW inVia Raman spectrometer. The thermostability of the coatings was measured with a thermogravimetric analyzer (TG, 209 F3). The surface hydrophobicity of materials was studied by a contact-angle-measuring instrument (POWEREACH). Characterization of the film structure was achieved using infrared spectroscopy (PerkinElmer, Spectrum Two). Electrochemical measurements were still carried out by a conventional three-electrode system described above, where the 316SS steel plate with or without coating was used as the working electrode. To quickly evaluate the anti-corrosion performance of the composite coating, the highly corrosive solution of the 0.1 M H_2_SO_4_ solution with 2 ppm HF was selected as the experimental solution to simulate the PEMFC environment.

The corrosion behaviors of the materials in the corrosive solution (0.1 M H_2_SO_4_ + 2 ppm HF, 70 °C) were evaluated using open circuit potential (Eocp), electrochemical impedance spectroscopy (EIS) and potentiodynamic polarization (PDP) tests. The interfacial contact resistance (ICR) values were measured to explore their conductivities according to the reference method [8].

Firstly, the sample was placed in a corrosive solution for 1 h. After, the exposed surface was fully immersed in water and the open circuit potential was stable. During the potentiodynamic polarization process, the potential was swept from −0.1 V to 0.6 V versus SCE with a scan rate of 1 mV/s after being immersed in the test solution for 1 h. The PDP test of 316SS was similar to that of the coatings, except that the scanning potential was replaced from −0.5 V to 0.4 V versus SCE.

To further investigate the stability of the polymer coatings in long-term immersion, electrochemical impedance tests were performed. Samples were placed in the corrosive solution and allowed to reach a stable open circuit potential (E_ocp_). Then, impedance tests were conducted at frequencies ranging from 0.1 Hz to 1 MHz, with an amplitude of 10 mV for the input sinusoidal voltage.

## 3. Results and Discussion

### 3.1. Characterization of Graphite Powders

A scanning electron microscope (SEM) was used to study the morphology of graphite. Appendix A shows that graphite powder is composed of a variety of large particles, with an estimating size distribution of 0.65–11.37 μm. The graphite powder was characterized by Raman spectroscopy. The ID/IG value was 0.907, indicating that a relatively high graphitization degree was found in the material. The carbon nitride nanosheets were tested by X-ray diffraction (XRD). As shown in Appendix A, the nanosheets were identified as g-C3N4, corresponding to (100) and (002) plane, respectively.

Using TGA, the thermal stability of the pPy and pPy/C coatings in an N2 atmosphere was studied. In Appendix A, it was evident that the pPy/C coatings had slightly improved thermal behavior in contrast to pure pPy before 200 °C. At the end of the analysis, the residues in the pPy and pPy/C coatings were 18.19% and 21.79% (*w*/*w*), respectively. Due to the encapsulation of graphite powder into a polymeric substrate, the pPy/C coating exhibited better thermal stability since the inorganic-oriented particles with rigid structures would inevitably reinforce the intrinsic nature of polymers.

The typical FT-IR spectra of the composite coatings in the range of 4000–400 cm^−1^ are shown in Appendix A. For comparison, the spectrum of graphite powder is also shown in Appendix A. From the FT-IR spectrum of CNNS, it can be seen that the strong and wide absorption peak at 3400–3100 cm^−1^ corresponds to the stretching vibration peak of the N-H bond, the peak at 1640–1230 cm^−1^ corresponds to the stretching vibration peak of the N-H bond, and the peak near 890 cm^−1^ corresponds to the tensile vibration peak of the N-H bond on the surface of triazine cyclic compounds. The peak near 806 cm^−1^ corresponds to the bending vibration peak of triazine cyclic compounds.

It can be seen from the FT-IR spectrum of PMMA that the peak at 2950 cm^−1^ corresponds to the stretching vibration peak of methylene, the absorption peak at 1730 cm^−1^ corresponds to the characteristic absorption peak of carbonyl C=O, the absorption peak at 1630 cm^−1^ corresponds to the characteristic absorption peak of PMMA, and the absorption peak at 1450 cm^−1^ corresponds to the characteristic absorption peak of methylene. The characteristic absorption peaks in CNNS and PMMA appeared in the infrared spectrum of the CNNS/PMMA film, respectively, which indicates the successful preparation of the CNNS/PMMA film. The absorption peaks at 3400 cm^−1^ correspond to the stretching vibration peaks of N-H bonds, and the absorption peaks at 1680 cm^−1^ and 1540 cm^−1^ correspond to the stretching vibration peaks of C-N bonds and C-C bonds in pyrrole. The absorption peaks at 1150 cm^−1^, 1030 cm^−1^ and 780 cm^−1^ correspond to the bending vibration peaks in plane and out of plane of the C-H bond. The infrared spectra of the CNNS/PMMA/pPy coating and the CNNS/PMMA/pPy-C coating combined the characteristic absorption peaks of the CNNS/PMMA coating and the pPy coating, indicating the successful preparation of a composite coating.

### 3.2. Electrodeposition of pPy-316 and pPy/C-316 Composite Coatings

Figure 1a refers to the current–time curves for pPy and pPy/C synthesized by 316SS in a 0.1 Mpy + 0.3 M oxalic acid solution at different deposition potentials. In Figure 1b, the current–time curve of pPy/C synthesized by 316SS in a 0.1 Mpy + 0.3 M oxalic acid + 1 g/L graphite powder suspension at different deposition potentials was recorded. In the process of electrodeposition, due to the electrical double layer, the current decreases at the beginning. Such an effect would be generally assigned to the passivation process of the 316SS surface. When pPy gradually nucleates on the surface of stainless steel, the current becomes more intensive and larger. As for the internal molecular structure of polypyrrole, such a polymeric framework contains numerous conjugated double bonds that would be composed of large quantities of electrons, which would be conducive to the migration and adsorption to the electrode. Compared with the 316SS surface, the pPy coating can induce the electrochemical polymerization of pyrrole on its surface, so the current increases. With the formation of the pPy coating, the current tends to be stable. It was found that as the current density increased, bubbles appeared on the surface of the pPy coating, indicating that an oxygen evolution reaction had occurred, and the FeC_2_O_4_·2H_2_O passivation film on the surface of 316SS began to dissolve, losing its protective effect on the metal.

### 3.3. The Anti-Corrosive Performance of Synthesized Coatings

With the aim of exploring the general effects, Figure 2 depicted the curves for an open circuit voltage versus time of bare 316SS and coated 316SS in the simulated PEMFC environment, respectively. In the process of soaking, corrosion occurs continuously, and the metal ions dissolved from the surface of stainless steel cover the surface of stainless steel to form a passive film, and the diffusion of metal cations was limited, which prevented the metal from further corrosion. Therefore, it was observed that the E_ocp_ of bare 316 stainless steel increased slowly, which indicated that there was a continuous corrosion process during long-term immersion in a corrosive environment. The E_ocp_ value of the CNNS/PMMA/pPy coatings and the CNNS/PMMA/pPy/C coatings decreased rapidly in the early stage, and then tended to be stable, because there were a large number of micropores on the coating surface, which made water molecules and corrosive ions quickly penetrate into the coating/metal matrix interface. After the open circuit potential was stable, the E_ocp_ value of coated stainless steel was much higher than that of 316 bare steel, which meant that the coatings had a better anti-corrosion effect. The distribution of micropores and microcracks on the surface of the coating would suppress the penetration and delay the entry path for the corrosion ions into the coating.

Potentiostatic polarization curves for bare 316SS and coated 316SS in the simulated PEMFC environment are provided in Appendix A. The curves for composite coatings showed that the current density dropped sharply at the beginning. Meanwhile, the curves for bare 316SS and a single coating are apparently identical. During the measurement time, the current density hardly changed.

In order to further explore the protection of the 316SS substrate by various coatings in a corrosive solution, the samples were recorded by electrochemical impedance spectroscopy. At the same time, in order to compare these results with the results of the unprotected stainless steel substrate, the EIS measurement of bare 316SS was also given. Figure 3 includes the Nyquist plots and Bode plots of bare 316SS, and the CNNS/PMMA, CNNS/PMMA/pPy and CNNS/PMMA/pPy/C coatings before and after soaking in a 0.1 M H2SO4 solution at 70 °C for 1 h.

Figure 3 shows the Nyquist plots and Bode plots of the sample in the simulated PEMFC working environment, and the experimental data were fitted by the equivalent circuit diagram. The derived fitting parameters are summarized in Appendix A. In the equivalent circuit diagram, Rs is the resistance of the solution, and CPE1 and Rcoat are related to the composite/electrolyte interface, where Rcoat represents the micropore resistance on the surface of the film material and CPE1 is the capacitance of the simulated coating material. CPE2 and Rct are related to the charge transfer reaction caused by electrolyte permeation. CPE2 is an electrical double-layer capacitance that simulates the bubble part of the interface, and Rct is attributed to the charge transfer resistance of the base metal corrosion reaction. The Rct in the fitting parameters is inversely proportional to the corrosion rate of the sample, indicating the corrosion rate. It can be seen from the fitting results that the fluctuation of Rs is small, indicating that the testing system is in a relatively stable state. Different from 316SS, the Nyquist plots of anti-corrosion coating consists of a capacitive arc with a large radius of curvature. Usually, the radius of capacitive arc reflects the size of charge transfer resistance during electrochemical corrosion, and the larger the radius of capacitive arc, the greater the charge transfer resistance and the better the corrosion resistance of the material. Analysis of the Bode plots showed that the coating has two time constants, one is the capacitor of the coating itself, and the other is the double-layer capacitor of the metal surface, indicating that the corrosion effect makes the coating change. pPy-0.90, pPy/C-0.80, and CNNS/PMMA/pPy/C-1.00 showed the largest |Z| values in the low-frequency region, with their maximum phase angles corresponding to the lowest frequency, where CNNS/PMMA/pPy/C-1.00 showed optimal corrosion resistance. In the frequency range from 1 to 100 Hz, a linear relationship was observed between the impedance and frequency, with a slope close to one.

Nyquist plots shows a small capacitance loop at high frequency, which indicates that the coating has relatively high conductivity at the beginning, so the surface of 316SS is in a passive state. Generally speaking, a high-frequency capacitive arc reflects the electrolyte/coating interface, while a low-frequency capacitive ring indicates the response of electrochemical process at the coating/316SS interface. According to Nyquist plots, the CNNS/PMMA/pPy/C-1.00 coating-synthesized constant voltage has the largest capacitance loop, which indicates that the coating has the largest resistance (Rct), the best mechanical isolation performance and great corrosion resistance. The increase tendency in the low-frequency region for samples mean that weak Warburg diffusion starts to appear, indicating that the metal has been corroded under the coating, the corrosion product film begins to affect the electrochemical reaction, and the coating has limited protection of the matrix. At the same time, for polypyrrole film layers without a CNNS/PMMA composite coating, pPy-0.90 and pPy/C-0.80 showed the largest capacity loop but not pPy-1.00 and pPy/C-1.00, and this may be related to the interaction between the CNNS/PMMA composite coating and the polypyrrole film layer. The Nyquist plots of 316SS and CNNS/PMMA-316 are similar, maybe indicating that the single CNNS/PMMA coating is insufficient to protect 316SS.

According to previous research, the anti-corrosion coating not only acts as a physical barrier to prevent the penetration of corrosive ions, but also provides anodic protection by the passive coating formed at the coating/metal interface [18]. The research shows that the pPy coating can improve the corrosion potential of the metal matrix, reduce the corrosion rate and better protect the stainless steel matrix from corrosion. In addition, the pPy coating can homogenize the metal potential, which means that the whole metal has a uniform potential. When the pPy coating on the metal surface becomes imperfect, the traditional coating often forms a small area of anode at the broken part, while the other protected parts can be regarded as the cathode. In this way, so called corrosive pitting at the broken part will occur. Due to the uniform potential of the pPy coating surface, corrosive pitting can be transformed into uniform corrosion. Therefore, it is of great significance to enhance the compactness of the coating by adding new functional moieties.

In order to obtain the corrosion effects of bare 316SS, CNNS/PMMA-, CNNS/PMMA/pPy- and CNNS/PMMA/pPy/C-coated stainless steel in a 0.1 M H_2_SO_4_ + 2 ppm HF solution, potentiodynamic polarization tests were performed (Figure 4). For bare 316SS, corrosive ions will erode the passivating coating on the surface of stainless steel. CNNS/PMMA was directly sprayed on the surface of 316 stainless steel, which would significantly slow down the corrosion process of the substrate to a certain extent. It was found that the coating obtained by spraying polypyrrole on 316SS in advance can be useful to improve stability. In Table 1, E_corr_ increases first and then decreases, while I_corr_ decreases first and then increases. According to the contents, CNNS/PMMA/pPy/C-1.00 has the largest E_corr_ value (109 mv), the smallest I_corr_ value (2.14 μA·cm^−2^), and the best corrosion resistance effect. Compared with bare 316SS, it was clear that the coating caused a positive shift of E_corr_ by more than 500 mV, and the I_corr_ value was decreased by more than an order of magnitude. The results verified that the CNNS/PMMA/pPy/C coating can effectively prevent corrosive substances from reaching the metal surface.

In Figure 5a, the surface morphology of bare 316SS was approximately smooth and a few scratches formed during mechanical polishing. According to Figure 5b, the distribution of the pPy/C coating was relatively uniform and dense. In essence, the whole structure was homogenous. Figure 5c presented the rough structure formed after spraying, with micropores and several nanoparticles on the coating surface. The deposited layer induced a more densely packed microstructure and the presence of tiny pores supported the formation of solvent volatilization after spraying.

To further investigate the elemental distribution and composition, EDX mapping of coated 316SS was conducted and presented in Appendix A. EDX mapping showed the homogeneous distribution of C, N, O and Fe elements.

The hydrophobicity of the coatings can be assessed macroscopically through the contact angle tests. Figure 6 showed the size of the contact angle of different interfaces. It can be seen that the contact angle of the pPy/C coating increases slightly compared with bare 316SS. After the incorporation of the sprayed coating, the contact angle becomes significantly larger. At the same time, a single CNNS/PMMA coating directly sprayed on the surface of bare 316SS significantly increases the contact angle compared with bare 316SS. This suggests that the CNNS/PMMA coating significantly improves the hydrophobicity of the coating, and the anti-corrosion coating shows a good physical barrier effect. Due to the insulation and hydrophobicity of the CNNS, its steric effect can form a physical barrier for the penetration of corrosive ions by adding it to the coating. In this manner, its diffusion channel in the coating would become narrow, thus enhancing the capability of the organic coating as the resistance to electrochemical corrosion.

We have also explored the interfacial contact resistance (ICR) values (Figure 7). We selected different samples synthesized at a potential of 1.00 V and bare 316SS. Appendix A demonstrated that the ICR values were in the range of 22–115 mΩ·cm^2^ upon exposure to a compaction pressure of 140 N/cm^2^ (this value is a typical loading force requirement for commercial PEMFCs), which were substantially larger than the DOE standard. The variation of ICR depends entirely on the internal structure and composition of the coating itself. Under a pressing pressure of 140 N·cm^−2^, the ICR of the pPy coating is 51 mΩ·cm^2^; compared to a single pPy coating, the ICR of the pPy/C coating is approximately 22 mΩ·cm^2^. Under the same pressure, the significant difference in ICR between the pPy and pPy/C coatings indicates that pPy/C coatings have better electrical conductivity than pPy coatings. This may be related to the excellent conductivity of graphite. The addition of CNNS/PMMA composite coatings significantly increases the ICR value due to the poor conductivity of the materials, which would ultimately gradually increase the electrical resistance.

## 4. Conclusions

Since the individual spacing layer could not achieve anti-corrosion effects in bipolar plates, robust and powerful hybrid protecting layers are expected. Herein, the pPy/C coating was electrodeposited on the surface of 316SS; and on this basis, a mixed solution of carbon nitride nanosheets and PMMA was sprayed. Electrochemical tests show that the samples with the pPy/C coating electrodeposited at 1.00 V potential in advance showed the best corrosion resistance. The CNNS/PMMA/pPy/C coating can be used as a functional layout to protect 316 stainless steel bipolar plates from proton-exchange membrane fuel cell corrosion, and prolong the service life of stainless steel bipolar plates.

## Figures and Tables

**Figure 1 molecules-28-03279-f001:**
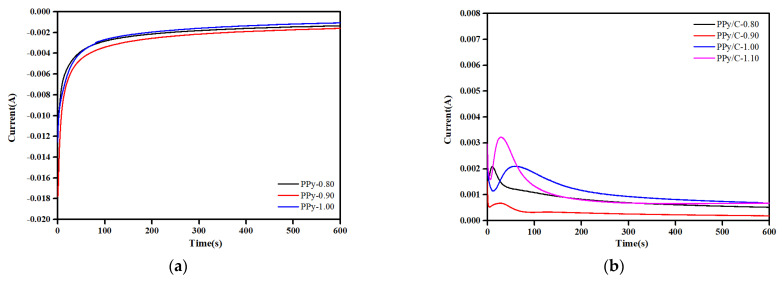
I–t curves for progress of synthesized coatings at different potentials in a magnetically stirred ice bath: (**a**) 0.1 Mpy + 0.3 M oxalic acid solution; (**b**) 0.1 Mpy + 1 g/L graphite powder + 0.3 M oxalic acid solution.

**Figure 2 molecules-28-03279-f002:**
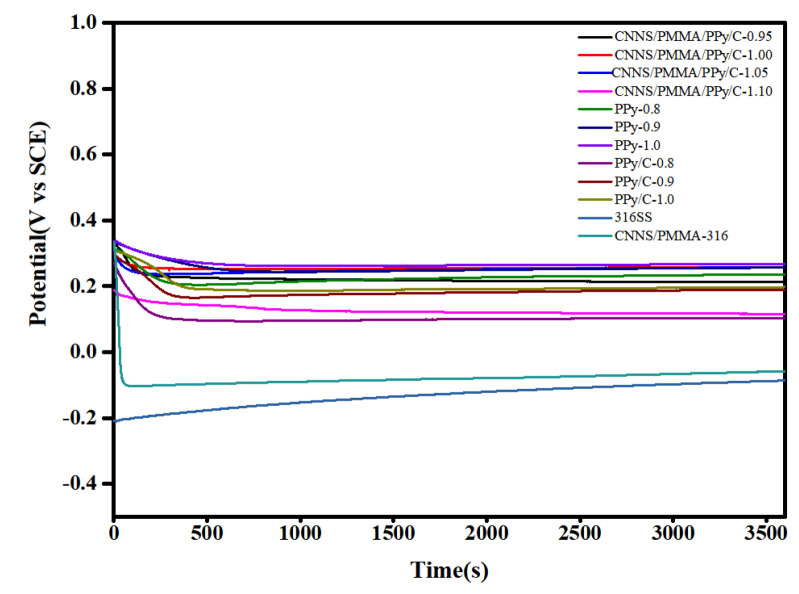
The E_ocp_ values versus time curves for bare 316SS and coated 316SS for 1 h immersion.

**Figure 3 molecules-28-03279-f003:**
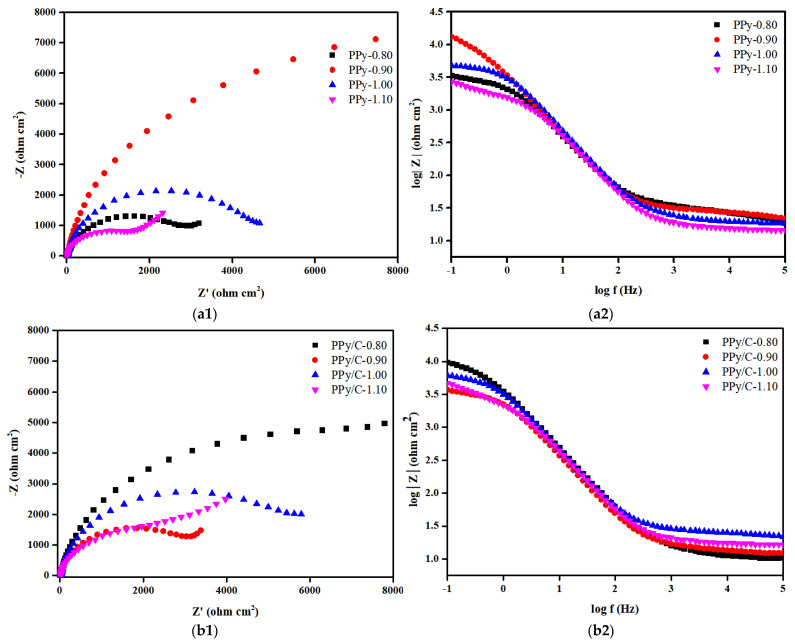
The Nyquist and Bode plots of: (**a**) pPy, (**b**) pPy/c, and (**c**) CNNS/PMMA/pPy/C.

**Figure 4 molecules-28-03279-f004:**
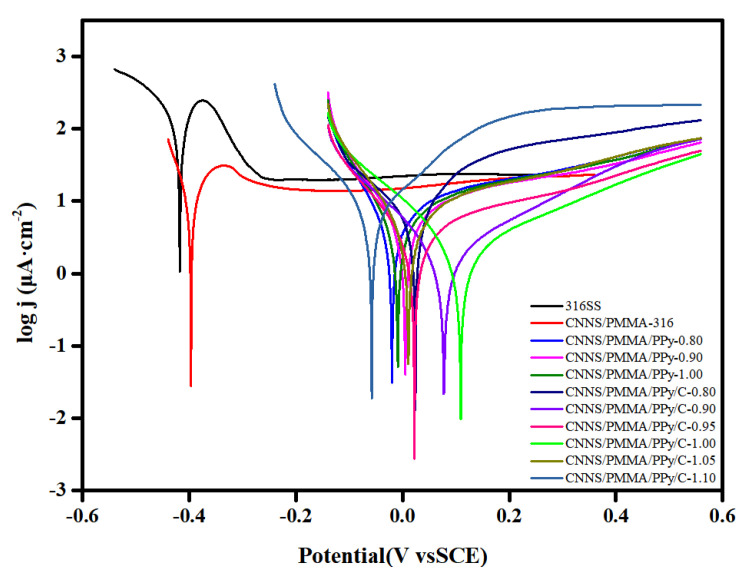
Potentiodynamic polarization of bare 316SS and coated 316SS.

**Figure 5 molecules-28-03279-f005:**
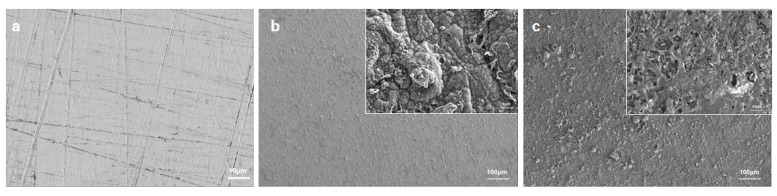
SEM images of different interfaces: (**a**) 316SS, (**b**) pPy/C, and (**c**) CNNS/PMMA/pPy/C.

**Figure 6 molecules-28-03279-f006:**
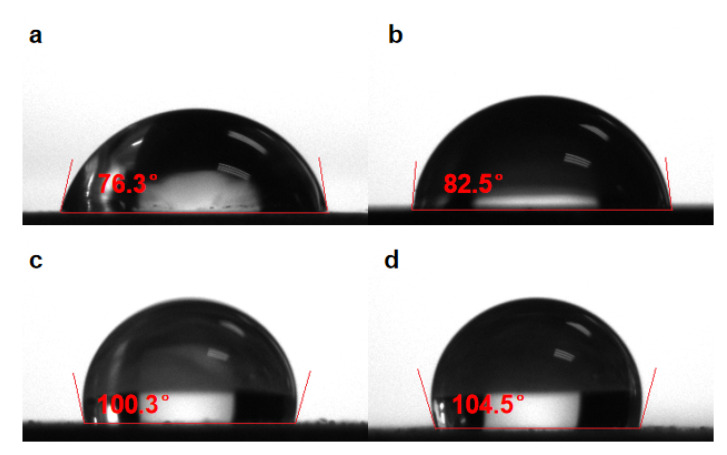
Contact angle of different interfaces: (**a**) 316SS, (**b**) pPy/C, (**c**) CNNS/PMMA/pPy/C, and (**d**) CNNS/PMMA.

**Figure 7 molecules-28-03279-f007:**
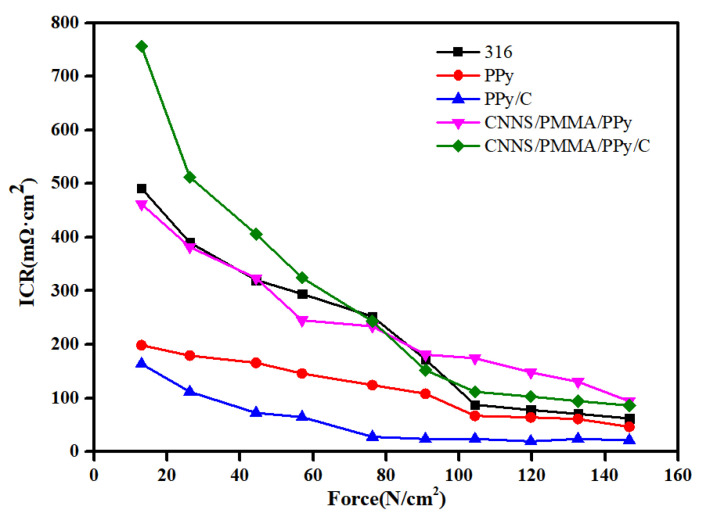
Contact angle of different interfaces: 316SS, pPy/C, CNNS/PMMA/pPy/C, and CNNS/PMMA. Interfacial contact resistance (ICR) of different samples.

**Table 1 molecules-28-03279-t001:** Fitting parameters of potentiodynamic polarization curves for different samples.

	E_corr_ (mV)	I_corr_ (μA·cm^−2^)
316SS	−418	46.45
CNNS/PMMA	−397	4.31
CNNS/PMMA/pPy-0.80	−20	4.58
CNNS/PMMA/pPy-0.90	5	5.45
CNNS/PMMA/pPy-1.0	−9	5.15
CNNS/PMMA/pPy/C-0.80	24	9.82
CNNS/PMMA/pPy/C-0.90	77	1.93
CNNS/PMMA/pPy/C-0.95	22	3.44
CNNS/PMMA/pPy/C-1.00	109	2.14
CNNS/PMMA/pPy/C-1.05	11	5.09
CNNS/PMMA/pPy/C-1.10	−58	9.22

## Data Availability

The data that support the findings of this study are available from the corresponding author upon reasonable request.

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
