# Peer review of "New Strategy for the Design of Anti-Corrosion Coatings in Bipolar Plates Based on Hybrid Organic–Inorganic Layers"

_molecules, 2023, doi:10.3390/molecules28073279_

Round 1
Reviewer 1 Report
I thank the Authors and the Molecules journal for the opportunity to read the manuscript “New strategy for the design of anti-corrosion coatings in bipolar plates based on hybrid organic-inorganic layers”, which, with some revision, will be a valuable contribution to the scientific literature. The research presented is indeed unique, yet some of the points require additional explanation and correction.
The hints and comments that may be of value to the Authors are provided below.
1) 1) The first question that arises from the reading of the manuscript: “Is carbon nitride nanosheets an accessible and affordable material that can be used commercially for modification of bipolar plates and what exactly is the role of this material in this research?”. Is the addition of carbon nitride nanosheets outweighs additional expenses?
2) 2)It is well known that PMMA is an insulating material, and hence, a composite coating based on this polymer might affect electrical conductivity for the worse. Additionally, both poly(methyl methacrylate) and polypyrrole have a low thermal conductivity that would also restrict the application of such coatings for bipolar plates. It is highly recommended to provide information about the electrical and thermal conductivity of the final coatings that comply with the target values set by the Department of Energy of the United States or other standards, since these characteristics impart fuel cell efficiency greatly.
3) 3)In the lines 216-219 the Authors explain that nucleation of the polypyrrole leads to an increase in the current, yet it is not clear how exactly these two phenomena are interconnected. Please, extend this explanation.
4) 4) In the lines 220-221 the Authors state that bubbles somehow indicate unsatisfactory bonding between steel and polymer coating, and again, this statement requires some extension.
5) 5) In Figure 2 it is not very clear why the open circuit potential of the coating with both polypyrrole and poly(methyl methacrylate) is lower than that of the coatings covered only with polypyrrole. It maybe that charge of the polypyrrole itself affected the results? Please, provide an additional explanation of this phenomenon.
6) 6)It would be much easier for the reader to understand the information presented in the manuscript if there were a table with all the designations of the coatings. Neither in the materials and methods, nor in the discussion section were clear details on the designation system used in the manuscript provided.
7) 7) In the Figure 3, a1 and b1 represent the Nyquist plots for the obtained coatings, however, Authors do not mention sharp increase in the low frequency region for ppy-1.1 and ppy-c-1.1, ppy-c-09. The Bode plots are not described as well. Please, extend explanation given.
8) 8) In the line 269 the Authors state that coatings obtained at a voltage of 1.0 V have the greatest capacitance loop, which is not quite in agreement with the plots presented in Figure 3. As it can be seen from the a1 and b1 figures, the greatest capacitance loops belong to coatings formed under voltages of 0.8 and 0.9 V (as far as these designations were properly understood). Please, extend figure captions or provide an additional explanation to the manuscript text.
9) 9) According to supplementary materials, corrosion current density undergoes a sharp decrease for CNNS/PMMA/ppy-c-0.9 in comparison with CNNS/PMMA/ppy-c-0.8 which was not explained in the manuscript. Please, elucidate the basis of this behavior of the coatings.
10) 10)For CNNS/PMMA/ppy-c-1.0 coating was stated that “the best corrosion resistance effect was achieved” which is not in the accordance with the information presented in Table S1 given in supplementary materials. It is also unclear from the table why corrosion parameters are presented twice. Please, extend the table captions.
11) 11)In the line 287 authors mention that the “Icorr value has been decreased by two orders of magnitude” for CNNS/PMMA/ppy-c-1.0 coating, yet it is not clear with what coating the comparison was made.
12) 12) In Figure 4 due to the overlapping of the curves presented, it is hard to assess the electrochemical properties of the obtained coatings without a table with calculated electrochemical properties, therefore, it is highly recommended to transfer this table from the supplementary materials to the main manuscript text.
13) 13) It would be very helpful to simulate the results of EIS with equivalent circuits in order to compare coatings resistance and examine their morphology.
14) 14) In the lines 313-314 the Authors mention “steric effect” which “can form a physical barrier for the penetration of corrosive ions”. Please elucidate what is meant by steric effect and how it can affect corrosive ions.
Author Response
Reviewer 1
Comments and Suggestions for Authors
I thank the Authors and the Molecules journal for the opportunity to read the manuscript “New strategy for the design of anti-corrosion coatings in bipolar plates based on hybrid organic-inorganic layers”, which, with some revision, will be a valuable contribution to the scientific literature. The research presented is indeed unique, yet some of the points require additional explanation and correction.
The hints and comments that may be of value to the Authors are provided below.
- The first question that arises from the reading of the manuscript: “Is carbon nitride nanosheets an accessible and affordable material that can be used commercially for modification of bipolar plates and what exactly is the role of this material in this research?”. Is the addition of carbon nitride nanosheets outweighs additional expenses?
Answer: Carbon nitride nanosheets could be easily prepared via the one-step roasting method based on simple precursor-melamine and the treatment process was easy to operate, and it can also meet the DOE standard (3 $KW-1). Due to its insulating and hydrophobic properties, CNNS can be added to the interior of the coating, and its steric resistance effect can set a barrier for the penetration of electrolyte and ions, making its diffusion channels within the coating curved and narrow, thereby enhancing the ability of organic coatings to resist electrochemical corrosion
- It is well known that PMMA is an insulating material, and hence, a composite coating based on this polymer might affect electrical conductivity for the worse. Additionally, both poly(methyl methacrylate) and polypyrrole have a low thermal conductivity that would also restrict the application of such coatings for bipolar plates. It is highly recommended to provide information about the electrical and thermal conductivity of the final coatings that comply with the target values set by the Department of Energy of the United States or other standards, since these characteristics impart fuel cell efficiency greatly.
Answer: Thanks for your useful suggestion, actually we have explored the interfacial contact resistance (ICR) values of CNNS/PMMA/PPy and CNNS/PMMA/PPy/C coated stainless-steel samples. Table S2 demonstrated that ICR values were measured to be in the range of 86-115 mΩ·cm2 upon exposure to the compaction pressure of 140 N/cm2, which were substantially larger than DOE standard (≤10 mΩ·cm2)..
The great conductivity of the coated anti-corrosion bipolar plate can improve the output power of the fuel cell, while the good corrosion resistance can prolong the service life of the battery. The dense coating can provide a good physical barrier, and the anode protection effect is related to the conductivity of the coating. When the coating becomes insulating, it only acts as a physical barrier. Therefore, the metal modified coating should achieve a reasonable match between conductivity and corrosion resistance.The mechanical strength of polypyrrole film layer is low, and its protection to metal substrate is limited. Preparing CNNS/PMMA coating on the surface not only enhances the corrosion resistance of the coating, but also improves its mechanical properties. At the same time, the hydrophobicity of the anticorrosive coating is also improved.
We have added the contents in page13, line 423-436.
Table S2 Interfacial contact resistance (ICR) values of CNNS/PMMA, CNNS/PMMA/PPy and CNNS/PMMA/PPy/C-1.00 coated stainless-steel samples at a compaction force of 140 N/cm2
Sample |
ICR (mΩ·cm2) |
316 |
66 |
PPy |
51 |
PPy/C |
22 |
CNNS/PMMA/PPy |
115 |
CNNS/PMMA/PPy/C |
86 |
- In the lines 216-219 the Authors explain that nucleation of the polypyrrole leads to an increase in the current, yet it is not clear how exactly these two phenomena are interconnected. Please, extend this explanation.
Answer: As for the internal molecular structure of polypyrrole, such polymeric framework contains numerous conjugated double bonds that would be composed of large quantities of electrons, which would conducive to the migration and adsorption to the electrode. We have added the contents in page6, line265-268.
Before electrodeposition, 316SS was completely exposed in electrolyte. When the voltage is applied, the current decreases rapidly due to the dissolution of 316SS, and a FeC2O4·2H2O passivation film is formed on the surface. Because of the high surface energy of passivation film, pyrrole can be induced to electrodeposit on the surface of passivation film. With the formation of polypyrrole film, the current decreases and tends to be stable. At the same time, because graphite has good conductivity, the current of electrodepositing PPy/C is larger than that of PPy after adding graphite powder.
- In the lines 220-221 the Authors state that bubbles somehow indicate unsatisfactory bonding between steel and polymer coating, and again, this statement requires some extension.
Answer: We have revised the sentence in page 6, line 258-260. Such PPy surface stared to lose its protection effects due to the increase of current density. As the current density increased, an oxygen evolution reaction had occurred, and the FeC2O4·2H2O passivation film on the surface of 316SS began to dissolve, losing its protective effect on the metal.
- In Figure 2 it is not very clear why the open circuit potential of the coating with both polypyrrole and poly(methyl methacrylate) is lower than that of the coatings covered only with polypyrrole. It maybe that charge of the polypyrrole itself affected the results? Please, provide an additional explanation of this phenomenon.
Answer: Thank you for your question, we have tried several times and the results supported the fact. It is estimated that partial oxidation of polypyrrole at open circuit potential might occur due to the possible reaction with nucleophile such as water. The results caused the degradation of the composite coating (polypyrrole and poly(methyl methacrylate)) without the assistance of graphite powder.
- It would be much easier for the reader to understand the information presented in the manuscript if there were a table with all the designations of the coatings. Neither in the materials and methods, nor in the discussion section were clear details on the designation system used in the manuscript provided.
Answer: Thank you for your suggestion, the synthesis method has been modified and the naming method has been detailed. We have revised the sentence in page 3, line 125-142 and page 4, line 164-168.
- In the Figure 3, a1 and b1 represent the Nyquist plots for the obtained coatings, however, Authors do not mention sharp increase in the low frequency region for ppy-1.1 and ppy-c-1.1, ppy-c-0.9. The Bode plots are not described as well. Please, extend explanation given.
Answer: The increase tendency in the low frequency region for samples mean that weak Warburg diffusion started to appear, indicating that the metal has been corroded under the coating, the corrosion product film began to affect the electrochemical reaction, and the coating has limited protection of the matrix. Derived from Bode plots, the impedance modulus has been selected to demonstrate the barrier performance. The results in general showed the slightly increased corrosion-protective behavior for all the coating samples compared with pure metal surfaces. Especially the sample of CNNS/PMMA/PPy/C-1.00 coating. Compared with other samples, it showed the largest |Z| values in the low frequency region, indicating the best corrosion resistance.
Relevant issues have been explained in page8-9, line314-350.
- In the line 269 the Authors state that coatings obtained at a voltage of 1.0 V have the greatest capacitance loop, which is not quite in agreement with the plots presented in Figure 3. As it can be seen from the a1 and b1 figures, the greatest capacitance loops belong to coatings formed under voltages of 0.8 and 0.9 V (as far as these designations were properly understood). Please, extend figure captions or provide an additional explanation to the manuscript text.
Answer: Thank you for your question, we mainly described CNNS/PMMA/PPy/C-1.00 which had the best corrosion resistance performance. The conjecture on the PPy-0.90 and PPy/C-0.80 have revised the sentence in page 9, line 350-353.
- According to supplementary materials, corrosion current density undergoes a sharp decrease for CNNS/PMMA/ppy-c-0.9 in comparison with CNNS/PMMA/ppy-c-0.8 which was not explained in the manuscript. Please, elucidate the basis of this behavior of the coatings.
Answer: Thank you for your useful question, the comparison of the order of magnitude of Icoor is not a significant multiple relationship. At the same time, due to the fact that the value of Icoor is mainly based on the tangent treatment of the electrokinetic polarization curve, there is a certain subjective error.
- For CNNS/PMMA/ppy-c-1.0 coating was stated that “the best corrosion resistance effect was achieved” which is not in the accordance with the information presented in Table S1 given in supplementary materials. It is also unclear from the table why corrosion parameters are presented twice. Please, extend the table captions.
Answer: Thank you for your question, the table has been recreated. The previous table actually described the PPy coatings and PPy/C coatings layers after spraying CNNS/PMMA composite coatings,
- In the line 287 authors mention that the “Icorr value has been decreased by two orders of magnitude” for CNNS/PMMA/ppy-c-1.0 coating, yet it is not clear with what coating the comparison was made.
Answer: Thank you for your question, the Icorr value of CNNS/PMMA/PPy/C-1.00 was comparised with bare 316SS. In this way, we could intuitively feel the protective effect of the coating on 316SS.
- In Figure 4 due to the overlapping of the curves presented, it is hard to assess the electrochemical properties of the obtained coatings without a table with calculated electrochemical properties, therefore, it is highly recommended to transfer this table from the supplementary materials to the main manuscript text.
Answer: We had recreated the table and transferred to the main manuscript text.
- It would be very helpful to simulate the results of EIS with equivalent circuits in order to compare coatings resistance and examine their morphology.
Answer: In the equivalent circuit diagram, Rs is the resistance of the solution, and CPE1 and Rcoat are related to the composite/electrolyte interface, where Rcoat represents the micropore resistance on the surface of the film material and CPE1 is the capacitance of the simulated coating material. CPE2 and Rct are related to the charge transfer reaction caused by electrolyte permeation. CPE2 is an electric double layer capacitance that simulates the bubble part of the interface, and Rct is attributed to the charge transfer resistance of the base metal corrosion reaction. The Rct in the fitting parameters is inversely proportional to the corrosion rate of the sample, indicating the corrosion rate. It was found that the resistance for charge transfer at the the CNNS/PMMA/PPy/C-1.00 coating interface was apparently larger than other samples, suggesting that the composite layers delayed the metal corrosion.
- In the lines 313-314 the Authors mention “steric effect” which “can form a physical barrier for the penetration of corrosive ions”. Please elucidate what is meant by steric effect and how it can affect corrosive ions.
Answer: Due to insulating and hydrophobic properties of CNNS, it can be added to the interior of the coating, and its steric resistance effect can set a barrier for the penetration of electrolyte and ions, making its diffusion channels within the coating curved and narrow, thereby enhancing the ability of organic coatings to resist electrochemical corrosion
Reviewer 2 Report
Review comments
In this paper, polypyrrole coating was prepared on the surface of 316 stainless steel by electrochemical method. On this basis, a composite coating composed of carbon nitride nanosheets (CNNS) and polymethyl methacrylate (PMMA) was sprayed. The morphology, wettability and corrosion resistance of the composite coating were characterized. The research content is substantial and has certain research significance, but there are still the following problems:
1. The purpose of this paper is to solve the problem of electrical conductivity and corrosion resistance of stainless steel bipolar plates. However, the sprayed polymethyl methacrylate and carbon nitride nanosheets are not conductive. Does spraying affect the conductivity of the coating? And the article lacks research related to coating conductivity.
2. In electrochemical impedance spectroscopy, low frequency capacitor ring and high frequency capacitor arc are very important parameters, which reflect different electrochemical processes and properties. Therefore, the analysis needs to be considered as a whole, not only part of the analysis. And the results of the low-frequency capacitor ring cannot be obtained that the coating impedance value prepared at 1 V is the largest. Please give a reasonable explanation.
3. The morphology and corrosion resistance of the coating were researched in this article, but the composition of the coating was not characterized. It is suggested to supplement relevant tests (such as Fourier infrared spectroscopy or X-ray electron spectroscopy).
4. In this paper, it is mentioned that polypyrrole coating can make the potential of metal matrix uniform. Does it inhibit corrosion or promote corrosion? There is no clear expression in the article. I hope the author can make further explanation.
5. Whether the description of the content of Figure 1 (a) in the article is accurate? Figure 1 (a) is the current-time curve after the synthesis of PPy on the surface of 316SS, excluding PPy/C coating. There is also a problem with the legend in Figure 1 (a), which should be 1.0 V instead of 0.10 V. The polarization curve is Figure 4 in the text, not Figure 4S in the support information. In addition, the words of 316 stainless steel in the text needs to be unified (316ss, 316 stainless steel, stainless steel).
6. The impedance spectra of blank stainless steel and surface directly sprayed CNNS/PMMA coated stainless steel are not given in Fig.3, but they are described in detail in this paper. Please add relevant data.
7. From the table of supporting information, it can be seen that compared with blank stainless steel, the corrosion current density of the coated sample decreases by only one order of magnitude, while it is indicated in the paper that the decrease is two orders of magnitude. Whether the statement is accurate?
8. The format of references needs to be consistent.
Author Response
Reviewer 2
Review comments
In this paper, polypyrrole coating was prepared on the surface of 316 stainless steel by electrochemical method. On this basis, a composite coating composed of carbon nitride nanosheets (CNNS) and polymethyl methacrylate (PMMA) was sprayed. The morphology, wettability and corrosion resistance of the composite coating were characterized. The research content is substantial and has certain research significance, but there are still the following problems:
1.The purpose of this paper is to solve the problem of electrical conductivity and corrosion resistance of stainless steel bipolar plates. However, the sprayed polymethyl methacrylate and carbon nitride nanosheets are not conductive. Does spraying affect the conductivity of the coating? And the article lacks research related to coating conductivity.
Answer: The great conductivity of the coated anti-corrosion bipolar plate can improve the output power of the fuel cell, while the good corrosion resistance can prolong the service life of the battery. The dense coating can provide a good physical barrier, and the anode protection effect is related to the conductivity of the coating. When the coating becomes insulating, it only acts as a physical barrier. Therefore, the metal modified coating should achieve a reasonable match between conductivity and corrosion resistance. The mechanical strength of polypyrrole film layer is low, and its protection to metal substrate is limited. Preparing CNNS/PMMA coating on the surface not only enhances the corrosion resistance of the coating, but also improves its mechanical properties. At the same time, the hydrophobicity of the anticorrosive coating is also improved.
- In electrochemical impedance spectroscopy, low frequency capacitor ring and high frequency capacitor arc are very important parameters, which reflect different electrochemical processes and properties. Therefore, the analysis needs to be considered as a whole, not only part of the analysis. And the results of the low-frequency capacitor ring cannot be obtained that the coating impedance value prepared at 1 V is the largest. Please give a reasonable explanation.
Answer: Relevant issues have been explained in page8-9, line314-329.
The Rct in the fitting parameters is inversely proportional to the corrosion rate of the sample, indicating the corrosion rate. It can be seen from the fitting results that the fluctuation of Rs is small, indicating that the testing system is in a relatively stable state. It was found that the resistance for charge transfer at the CNNS/PMMA/PPy/C-1.00 coating interface was apparently larger than other samples, suggesting that the composite layers delayed the metal corrosion.
- The morphology and corrosion resistance of the coating were researched in this article, but the composition of the coating was not characterized. It is suggested to supplement relevant tests (such as Fourier infrared spectroscopy or X-ray electron spectroscopy).
Answer: The relevant data has been supplemented. We have added the FT-IR spectra in page 5, line 226-248. At the same time, we added the EDX-mapping in supporting as Fig. S7 and the contents in page11, line 401-403.
- In this paper, it is mentioned that polypyrrole coating can make the potential of metal matrix uniform. Does it inhibit corrosion or promote corrosion? There is no clear expression in the article. I hope the author can make further explanation.
Answer: The previous research shows that PPy coating can improve the corrosion potential of metal matrix, reduce the corrosion rate and better protect stainless steel matrix from corrosion. In addition, PPy coating can homogenize the metal potential, which means that the whole metal has a uniform potential. When the PPy coating on the metal surface becomes imperfect, the traditional coating often forms a small area of anode at the broken part, while the other protected parts can be regarded as cathode. In this way, the so-called corrosive pitting at the broken part will occur. Due to the uniform potential of PPy coating surface, corrosive pitting corrosion can be transformed into uniform corrosion.
- Whether the description of the content of Figure 1 (a) in the article is accurate? Figure 1 (a) is the current-time curve after the synthesis of PPy on the surface of 316SS, excluding PPy/C coating. There is also a problem with the legend in Figure 1 (a), which should be 1.0 V instead of 0.10 V. The polarization curve is Figure 4 in the text, not Figure 4S in the support inf-rmation. In addition, the words of 316 stainless steel in the text needs to be unified (316ss, 316 stainless steel, stainless steel).
Answer: The words of 316 stainless steel have been unified as 316 and the relevant content has been modified.
- The impedance spectra of blank stainless steel and surface directly sprayed CNNS/PMMA coated stainless steel are not given in Fig.3, but they are described in detail in this paper. Please add relevant data.
Answer: Thank you very much for your reminder, we had added relevant data in Fig.3.
- From the table of supporting information, it can be seen that compared with blank stainless steel, the corrosion current density of the coated sample decreases by only one order of magnitude, while it is indicated in the paper that the decrease is two orders of magnitude. Whether the statement is accurate?
Answer: The specific description has been modified. Thank you for your amicable suggestion.
- The format of references needs to be consistent.
Answer: The references have recited and revised.
Reviewer 3 Report
The authors demonstrate the polypyrrole/graphite coating on the surface of 316 stainless steel by potentiostatic method, then mixed carbon nitride nanosheets and PMMA solution, and sprayed the mixed solution on the surface of polypyrrole/graphite coating to prepare corrosion-resistant coatings. The corrosion resistance of these coatings in the simulated proton exchange membrane fuel cell environment has been studied by electrochemical test. The manuscript is well-defined; however, at certain points manuscript should be reconsidered and revised.
1. Introduction is too long must be concise.
2. Last para of introduction is similar to abstract. It must include the strategies applied to solve the challenges. while abstract mention the important outcomes of this study.
3. section 3.1 materials and reagents. Chemicals used are missing in order to repeat the experiment, upon readers choice.
Author Response
Reviewer 3
The authors demonstrate the polypyrrole/graphite coating on the surface of 316 stainless steel by potentiostatic method, then mixed carbon nitride nanosheets and PMMA solution, and sprayed the mixed solution on the surface of polypyrrole/graphite coating to prepare corrosion-resistant coatings. The corrosion resistance of these coatings in the simulated proton exchange membrane fuel cell environment has been studied by electrochemical test. The manuscript is well-defined; however, at certain points manuscript should be reconsidered and revised.
- Introduction is too long must be concise.
Answer: Modifications for the introduction section have been made.
- Last para of introduction is similar to abstract. It must include the strategies applied to solve the challenges. while abstract mention the important outcomes of this study.
Answer: Thank you for your suggestion, we had revised the last section of introduction, which has been given in page 3.
3.section 3.1 materials and reagents. Chemicals used are missing in order to repeat the experiment, upon readers choice.
Answer: The specific reagents have been modified in page 3, line 113-118.
Round 2
Reviewer 1 Report
The Authors have provided comprehensive answers and have added all necessary corrections to the text; the presented version of the manuscript is ready to be published.